# Recombinant SAG2A Protein from *Toxoplasma gondii* Modulates Immune Profile and Induces Metabolic Changes Associated with Reduced Tachyzoite Infection in Peritoneal Exudate Cells from Susceptible C57BL/6 Mice

**DOI:** 10.3390/microorganisms12112366

**Published:** 2024-11-20

**Authors:** Thaíse Anne Rocha dos Santos, Mário Cézar de Oliveira, Edson Mario de Andrade Silva, Uener Ribeiro dos Santos, Monaliza Macêdo Ferreira, Ana Luísa Corrêa Soares, Neide Maria Silva, Tiago Antônio de Oliveira Mendes, Jamilly Azevedo Leal-Sena, Jair Pereira da Cunha-Júnior, Tiago Wilson Patriarca Mineo, José Roberto Mineo, Érica Araújo Mendes, Jane Lima-Santos, Carlos Priminho Pirovani

**Affiliations:** 1Centro de Biotecnologia e Genética, Universidade Estadual de Santa Cruz, Ilhéus 45662-900, BA, Brazil; tayseesantos@gmail.com (T.A.R.d.S.); monalizamacedo2@gmail.com (M.M.F.); millybio7@gmail.com (J.A.L.-S.); pirovani@uesc.br (C.P.P.); 2Instituto de Ciências Biomédicas, Universidade Federal de Uberlândia, Uberlândia 38408-100, MG, Brazil; cezar_cle@yahoo.com.br (M.C.d.O.); nmsilva@ufu.br (N.M.S.); jair.cunha.junior@gmail.com (J.P.d.C.-J.); tiago.mineo@ufu.br (T.W.P.M.); jrmineo@ufu.br (J.R.M.); 3Departamento de Bioquímica e Biologia Molecular, Universidade Federal de Viçosa, Viçosa 36570-900, MG, Brazil; mariodeandradee@gmail.com (E.M.d.A.S.); tiagoaomendes@ufv.br (T.A.d.O.M.); 4Laboratório de Imunobiologia, Universidade Estadual de Santa Cruz, Ilhéus 45662-900, BA, Brazil; uener_edu@yahoo.com.br; 5Laboratory of Tropical Crop Improvement, Division of Crop Biotechnics, KU Leuven, 3000 Leuven, Belgium; anacorreasoares@outlook.com; 6Departamento de Microbiologia, Universidade de São Paulo, São Paulo 05508-000, SP, Brazil; ericaarmendes@gmail.com

**Keywords:** *Toxoplasma gondii*, macrophages, rSAG2A, cytokine, inflammation

## Abstract

Toxoplasmosis is a neglected disease that represents a significant public health problem. The antigenic profile of *T. gondii* is complex, and the immune response can lead to either susceptibility or resistance. Some antigens, such as surface antigen glycoprotein (SAG), are expressed on the surface of tachyzoite stages and interact with the host immune cells. In this study, we investigated the potential of the recombinant SAG2A protein of *T. gondii* to control parasitism and modulate the immune response in the peritoneal exudate cells (PECs) of both susceptible (C57BL/6) and resistant (BALB/c) mice using an in vitro infection model, gene expression, proteomic analysis, and bioinformatic tools. Our results showed that rSAG2A-treated PECs presented a lower parasitism in C57BL/6 mice but not in the PECs from BALB/c mice, and induced a pro-inflammatory cytokine profile in C57BL/6 mice (*iNOS*, *TNF-α*, and *IL-6*). rSAG2A modulated different exclusive proteins in each mouse lineage, with PECs from the C57BL/6 mice being more sensitive to modulation by rSAG2A. Additionally, biological processes crucial to parasite survival and immune response were modulated by rSAG2A in the C57BL/6 PECs, including fatty acid beta-oxidation, reactive oxygen species metabolism, interferon production, and cytokine-mediated signaling pathways. Together, our study indicates that rSAG2A controls *T. gondii* parasitism in susceptible C57BL/6 PECs through the modulation of pro-inflammatory cytokines and enhanced expression of proteins involved in the cytotoxic response.

## 1. Introduction

Toxoplasmosis is a neglected zoonotic disease caused by *Toxoplasma gondii*, which can infect a wide range of warm-blooded vertebrates [1]. Approximately one-third of the world’s population has antibodies against *T. gondii*, and in some European countries, around 80% of the population have been exposed to the parasite at some point in their lives. In Brazil, recent years have been marked by several outbreaks of Toxoplasmosis in different states [2,3,4]. The disease can be fatal in congenital infections and can also cause cranio-cerebral and ocular sequelae in surviving neonates. In fact, infections in immunosuppressed patients and pregnant woman infected during the first trimester of pregnancy present the highest risk [5]. 

The life cycle of the parasite is heteroxenous, with felines serving as the definitive hosts. When infected, felines shed millions of oocysts per day in their feces. These oocysts sporulate and become infective in the environment. Intermediate hosts likely include all warm-blooded animals, such as most livestock and humans. Hosts become infected by ingesting sporulated oocysts that contaminate crops, soil, and water sources, or by consuming raw or undercooked meat containing cysts, which are the two primary horizontal transmission routes [6]. In the intestinal cells, these forms of the parasite differentiate into tachyzoites, which rapidly replicate in acute-disease-causing forms. As the immune response develops, some tachyzoites escape destruction and develop into bradyzoites, which form cysts in various tissues, including the brain and skeletal muscle. In immunocompetent hosts, these cysts do not cause overt disease and remain undetected as a relatively benign chronic infection. However, in immunocompromised patients, latent infection can reactivate, with bradyzoites converting into rapidly replicating tachyzoites, causing severe, life-threatening disease [6,7]. In pregnant women, acute infections acquired during or shortly after gestation can lead to congenital toxoplasmosis.

*T. gondii* has a complex antigenic profile that changes over the parasite’s life cycle, including tachyzoite, bradyzoite, and sporozoite stages. These antigens include micronem (MIC), apical membrane antigens (AMAs), rhomboid antigens (ROMs), rhoptry antigens (ROPs), dense granule (GRAs) antigens, and surface antigen glycoproteins (SAGs) [8]. However, the immune response that these antigens promote and their immunogenicity vary, making effective vaccine development challenging [8]. The immune response to *T. gondii* can lead to different immunological outcomes, susceptibility, and resistance. Once *T. gondii* cross the mucosal barrier in the hosts, they activate immune cells to produce tumor necrosis factor-alpha (TNF-α) and anti-inflammatory cytokines such as interleukin 10 (IL-10), IL-27, and transforming growth factor-beta (TGF-β) [9]. This downregulates pro-inflammatory cytokines, enhances parasite proliferation, and triggers parasite migration to immune-privileged sites, including the brain, eye, and placenta. Conversely, an immune response that activates cell-mediated immunity and interferon-gamma (IFN-γ) production by natural killer cells and CD8^+^ T cells controls *T. gondii* infection [9]. Neutrophils, macrophages, and dendritic cells that recognize the parasite evoke a pro-inflammatory response with high levels of IL-1β, IL-12, IL-18, and TNF-α, inhibiting parasite proliferation and infection [9]. 

Interestingly, different *T. gondii* antigens control the immune response. For instance, IL-22 is produced by CD4^+^ T_H_17 cells in response to the presence of ROP antigens [10,11]. GRAs enhance interferon regulatory factor 8 (IRF8), nuclear factor kappa B (NF-κB), and T-bet, controlling the CD4^+^ T_H_1/T_H_2 immune response [12]. Additionally, SAG antigens can induce INF-γ and IL-12 production against *T. gondii* [13].

SAG2A is a protein of 22 kDa that belongs to the SAG2-like sequence family [14]. It differs from the other SAG proteins due to the presence of a disordered structure in its C-terminal region. SAG2A is an immunodominant antigen expressed in the tachyzoite phase of *T. gondii*, which contains a B cell epitope, making it a promising candidate for vaccine and diagnostic purposes [14,15,16]. SAG2A interacts with the host immune cells during acute infection, as it is abundantly expressed on the surface of tachyzoites [17]. The detection of IgG anti-SAG2A has been proposed for the diagnosis of toxoplasmosis. The use of IgG1 anti-SAG2A [15] and IgG3 anti-recombinant SAG2A [16] has shown promising results and a higher sensitivity in the acute phase of disease. Interestingly, these authors observed that the determination of the IgG3/IgG1 ratio of antibodies specific for rSAG2A associated with classic serum markers could be a tool to distinguish the early acute phase of *T. gondii*-infected patients.

The protein SAG2A is released from *T. gondii* tachyzoites in gliding assays using “freshly harvested” *T. gondii* RH strain tachyzoites [18]. Additionally, murine innate immune cells can be modulated by SAG2A [14]. These results suggest that SAG2A can be released during the beginning of infection, while also being able to interact with compounds from the adaptive immune response, inducing a robust humoral immune response during the initial phases of infection. To understand the role of SAG2A in the parasite immune response in toxoplasmosis, we evaluated the impact of the recombinant SAG2A protein (rSAG2A) during the infection of peritoneal exudate cells (PECs) from BALB/c and C57BL/6 mice in comparison with untreated infected macrophages. Our results showed that rSAG2A-primed PECs presented a lower number of intracellular tachyzoites in susceptible C57BL/6 mice, but not in PECs from resistant BALB/c mice. We characterized the immunological profile in both lineages and analyzed the potential metabolic pathways involved in the reduced parasitism of PECs from C57BL/6 mice infected with *T. gondii* compared to PECs treated with rSAG2A.

## 2. Materials and Methods

### 2.1. Ethics Statement

BALB/c (*n* = 15) and C57BL/6 (*n* = 15) male mice (5–8 weeks) were purchased from the Federal University of Uberlândia Rodent Bioterium and maintained in standard conditions, with a 12/12 h light/dark cycle under specific pathogen-free conditions (SPF) and with ad libitum water and chow. All the experiments with mice were carried out according to the institutional guidelines and were approved by the ethics institutional care and use committee of the State University of Santa Cruz, under protocol number 004/11.

### 2.2. Parasites

*T. gondii* tachyzoites RH strain (2F1 clone), which constitutively expresses cytoplasmic β-galactosidase [19], was kindly donated by Dr. Vern Carruthers of the University of Michigan Medical School (Michigan, MI, USA). The tachyzoites were propagated in human cervix adenocarcinoma (HeLa) cell lines obtained from the American Type Culture Collection (ATCC, Manassas, VA, USA) and cultured in Roswell Park Memorial Institute (RPMI) 1640 medium (Cultilab, Campinas, SP, Brazil) supplemented with L-glutamine, penicillin, streptomycin, and 2% fetal bovine serum (Cultilab) at 37 °C and 5% CO_2_. 

### 2.3. Isolation of Peritoenal Exudate Cells (PECs)

The animals received an intraperitoneal injection of 2 mL of 3% thioglycolate medium, and after 72 h, the peritoneal exudate cells (PECs) rich in peritoneal macrophages were collected in 5 mL of PBS after euthanasia, as previously suggested [20]. The cells were washed, counted, and suspended in DMEM medium (Cultilab) supplemented with 10% of fetal bovine serum (FBS) + 40 mg mL^−1^ of gentamicin (Gibco by Invitrogen, Waltham, MA, USA). 

### 2.4. Parasitism and SAG2A 

*T. gondii* parasitism in the PECs was measured using the β-Gal assay, as previously suggested [21]. Two independent experiments were performed using a pool of PECs obtained from *n* = 4 animals in each experiment. The PECs of BALB/c and C57BL/6 mice were incubated (1 × 10^5^ cells/200 μL/well) in 10% DMEM medium in 96-well plates for 24 h at 37 °C and 5% CO_2_. After the incubation period, the cells were washed to remove non-adherent cells, infected or not with tachyzoites from 2F1 clone in a 1:1 parasites/cell proportion (multiplicity of infection, MOI = 1). Three hours after the infection, the cell monolayers were rinsed with fresh DMEM to remove non-adherent and extracellular parasites, followed by the addition of 10 μg mL^−1^ of the full-length rSAG2A protein. The full-length rSAG2A was obtained as previously described [22]. The dose concentration (10 μg/mL^−1^) used presented no toxicity to the PECs from both BALB/c and C57BL/6 mice, as previously described [22]. After 24 h, the parasitism was quantified using chlorophenol red–β-D-galactopyranoside (CPRG; Roche, Mannheim, Germany), as previously described [21]. The β- galactosidase activity was measured at 570 nm in a microplate reader (Versa Max ELISA Microplate Reader, Molecular Devices, Sunnyvale, CA, USA).

### 2.5. RT-qPCR

The PECs (1 × 10^6^ cells) from BALB/c and C57BL/5 mice were placed in 24-well plates, infected with 2 × 10^6^ tachyzoites, and incubated for 3 h at 37 °C and 5% CO_2_. After that, the cells were treated or not with 10 μg/mL^−1^ of rSAG2A protein for 24 h. The samples were collected in 500 μL of TRIzol (Invitrogen, Waltham, MA, USA), followed by RNA extraction according to the manufacturer’s protocol. The RNA concentration was determined using a NanoDrop 2000 spectrophotometer (Thermo Fisher Scientific, Waltham, MA, USA) after RNA treatment with DNAse I (Thermo Fisher Scientific), and the RNA integrity was evaluated using 1% agarose gel. cDNA synthesis was performed using the cDNA First Strand kit (Thermo Fisher Scientific), according to the manufacturer’s instructions, followed by quantification and sample dilution to achieve 40 ng/1 μL. Amplification was carried out with Maxima SYBR Green/GoTaq qPCR Master Mix (Promega, Madison, WI, USA) using a Stratagene Mx3005P system (Agilent Technologies, Santa Clara, CA, USA). The gene markers included *IL-1β*, *IL-6*, *IL-10*, *TGF-β*, *TNF-α*, *Arginase 1*, and *iNOS*, and the gene *18S* was used for normalization (Table 1). The PCR conditions were as follows: 95 °C for 10 min and 40 cycles of 95 °C for 5 s, 60 °C for 30 s, and 75 °C during 30 s. The reaction products were dissociated to confirm whether the products obtained were unique and specific. The relative expression was measured using the 2^−ΔΔCt^ methodology [23]. The mean values of 2^−ΔΔCt^ for each gene were converted using a Z-score scale and these values were used for a heatmap analysis with clusterization. These analyses were implemented in the R environment (3.6.2) with the Heatmap function, using the ComplexHeatmap package [24,25]. Grouping was performed using the mean Euclidean distance and the complete method.

### 2.6. Quantitative Proteomic

A pool of PECs obtained from n = 7 male BALB/c and C57BL/6 mice were plated in 6-well plates in triplicates using 5 × 10^6^ cells/well. Three conditions were established: a control group with PECs without infection and not treated with rSAG2A, PECs infected with *T. gondii* (MOI = 1), and PECs treated with rSAG2A (10 µg/mL^−1^). The cells were incubated in a BOD incubator for 24 h at 37 °C and 5% CO_2_. Protein extraction was carried out according to the phenol method, as previously described [26]. The cells were treated with 750 µL of extraction buffer (100 mmol L^−1^ Tris-HCL pH 8.3; 5 mmol L^−1^ EDTA; 100 mmol L^−1^ KCL; 1% p/v DTT; 30% sucrose; 1 complete Mini EDTA-free protease inhibitor cocktail tablet (Roche Applied Science, Penzberg, Upper Bavaria, Germany) per 10 mL of buffer. The samples were then vortexed and centrifuged at 12,000 rpm at 4 °C. The phenolic phase was collected, followed by the addition of 750 µL of buffered phenol, vortexing, and centrifugation for five minutes. The phenolic phase was collected once more, added with 100 mmol/L of ammonium acetate in methanol and left overnight at −20 °C to precipitate. After that, the samples were centrifuged for 60 min at 4 °C and 13,000 rpm, the supernatant was discarded and the pellet was kept for 60 min in cold acetone and 0.2% DTT solution at −20 °C and rinsed twice in the same solution. Between the rinsing processes, the samples were centrifuged (4 °C and 13,000 rpm) and the supernatants were discarded. The pellet was gently dried under a hood at room temperature and resuspended with 200 µL of lysis buffer (8 mol/L urea; 5 mmol/L DTT; 30 mmol/L Tris). The protein concentration was quantified by the Bradford quantification method and samples were lyophilized and sent to the Proteomics Lab of the Faculty of Bioscience Engineering and SyBioMa of the Biomedical Sciences Group, KU Leuven, for further proteomics workflow steps. For protein digestion, the samples were re-suspended in 200 μL of lysis buffer and the following process was performed successively, adding the following compounds for each 20 µg of proteins: 20 mmol/L of DTT and 15 min incubation; 50 mmol/L of iodoacetamide, incubation in the dark for 30 min; three times dilution with 150 mmol/L of ammonium bicarbonate; the addition of 0.2 μg of trypsin and overnight incubation at 37 °C; and trifluoracetic acid to a 0.1% final acidification concentration. After that, Pierce™ C18 Spin Columns (Thermo Fisher Scientific) were used for the desalting of the samples, according to the manufacturer’s instructions. Peptides separation of the digested samples (1 µg/5 µL^−1^) was performed via a UPLC-MS/MS system, with the Ultimate 3000 UPLC system (Dionex, ThermoScientific, Waltham, MA, USA) and Q Exactive Orbitrap mass spectrometer (Thermo Scientific) as previously described [27]. Data were obtained with the Xcalibur 3.0.63 software (ThermoScientific). 

### 2.7. Proteomic Data Analysis

Protein identification was performed with raw data conversion by Proteome Discover version 1.4 (Thermo Fisher Scientific) into mgf files and processing with MASCOT version 2.2.06 (Matrix Science, Columbus, OH, USA) against the Uniprot Mus musculus database (55,398 proteins). The false discovery rate (FDR) was calculated to enhance the identification confidence with Scaffold (version 3.6.3; Proteome Software Inc., Portland, OR, USA). Protein quantification was processed using Progenesis LC–MS version 4.1 (Nonlinear Dynamics, Milford, MA, USA), with automatic alignment from a selected reference run based on ion intensity.

Normalized fluorescence data obtained in the proteomic analysis of BALB/c and C57BL/6 mice PECs in the control condition without rSAG2A and not infected with *T. gondii*; cells only infected with the parasite; and cells only treated with rSAG2A were used to calculate the fold change (treated/control). Proteins with a fold change (FC) higher than 1 were considered to be up-modulated (augmented) and proteins with FC < 1 were considered to be down-modulated (suppressed). These proteins were used to select those exclusively augmented or suppressed in the PECs of C57Bl/6 mice under the conditions of rSAG2A use or *T. gondii* infection.

### 2.8. Identification of Proteins Exclusively Up-Modulated in PECs of C57BL/6 Mice Treated with rSAG2A

To identify exclusively augmented proteins in the PECs from C57BL/6 mice, a qualitative approach was used. Briefly, the FC was calculated based on comparisons between the treated (rSAG2A) and infected (*T. gondii*) groups with their respective controls: (i) BALB/c_rSAG2A/Control; (ii) BALB/c_*T. gondii*/Control; (iii) C57BL/6_rSAG2A/Control; and (iv) C57BL/6_*T. gondii*/Control. When a protein displayed an FC > 1 exclusively in a determined condition, it was considered as exclusive for the specific condition. The proteins with an FC > 1 for two or more conditions were considered to be shared proteins. Based on this principle, a Venn diagram was plotted using the bash/awk routine in the Linux system. Exclusive or up-modulated proteins present in the PECs from C57BL/6 mice treated with rSAG2A (FC > 2) were compared to the PECs of BALB/c mice treated with rSAG2A. Exclusive or up-modulated proteins present in the PECs from C57BL/6 mice infected with *T. gondii* (FC >2) compared to the PECs of BALB/c mice infected with *T. gondii* were highlighted in blue in the Venn diagram. 

We also selected down-modulated proteins (FC < 1.0) exclusively found in the PECs from C57BL/6 mice treated with rSAG2A and down-modulated proteins (FC < 1.0) exclusively found in the PECs from C57BL/6 mice infected with *T. gondii*. The fold changes of the selected proteins identified in Appendix A are demonstrated in the heatmap, plotted using the Heatmap function of the Complex Heatmap package in R version 4.3.3 [24,25].

### 2.9. Prediction of Protein-Protein Interaction Network

To understand what biological processes were regulated in the PECs of C57BL/6 and BALB/c mice treated with the rSAG2A protein, we conducted a study of protein–protein interactions. We focused on groups of exclusive or up-modulated proteins and on down-modulated proteins due to rSAG2A or infection by *T. gondii*. In this sense, two networks were generated for the PECs of C57BL/6 mice treated with rSAG2A: (i) a network of exclusive proteins or those with augmented accumulation when the PECs were treated with rSAG2A; and (ii) a network for the group of down-modulated proteins. Two other networks were generated for the PECs of C57BL/6 mice infected with *T. gondii*: (i) a network of exclusive or up-modulated proteins when the PECs were infected with *T. gondii*; and (ii) a network of proteins down-modulated when the PECs were infected with *T. gondii*. These networks were generated starting from a high-reliability network (0.700) of the Mus musculus interactome in the STRING database [28]. Module and gene ontology analyses were carried out as described by those authors. For these analyses, the plugins from the Cytoscape MCODE [29] and BiNGO (Biological Network Gene Ontology) [30] were used, respectively. To conduct the gene ontology, a notation file was obtained using Ensemble. 

To understand which proteins modulated by rSAG2A were more associated with the inflammatory response, we constructed protein-protein interaction networks starting from the cytokines whose gene expression profile was evaluated in the present study. The proteins modulated by rSAG2A were highlighted and the functional enrichment of the biological processes of the main groups were again analyzed with BiNGO.

### 2.10. Statistical Analysis

Data were analyzed using GraphPad Prism version 5.0 (GraphPad Software, San Diego, CA, USA). Statistical differences between groups were obtained through one-way ANOVA test, followed by Tukey or *t*-test. The results were considered significant with *p* < 0.5.

## 3. Results

### 3.1. rSAG2A Reduced T. gondii Parasitism in PECs from Susceptible C57BL/6 Mice, but Not in PECs from Resistant BALB/c Mice

To determine whether recombinant SAG2A modulates the immune response and impacts *T. gondii* infection, we employed an in vitro model using peritoneal exudate cells (PECs) from BALB/c and C57BL/6 mice. The rationale for using this model was the possibility of investigating the response of susceptible and resistant mice to *T. gondii* tachyzoite (2F1 clone) infection. We observed that, in early infection (24 h), PECs from both the resistant and susceptible mice showed similar levels of parasitism (*p* > 0.05). Interestingly, when treated with rSAG2A, the susceptible mice (C57BL/6) exhibited a significant reduction in parasitism (*p* < 0.001), with a decrease of 2000-fold compared to the untreated C57BL/6 mice (Figure 1). These data suggest that the rSAG2A protein can modulate the immune response in PECs to control *T. gondii* infection.

### 3.2. rSAG2A Reduced T. gondii Parasitism in PECs from Susceptible C57BL/6 Mice but Not in PECs from Resistant BALB/c Mice

To understand the effect of rSAG2A in modulating the immune response, we treated the PECs with rSAG2A in the presence of absence of *T. gondii* in the resistant and susceptible mice. First, we observed that both the resistant BALB/c and susceptible C57BL/6 mice exhibited the same basal immunological profile in control groups (Figure 2). Interestingly, the PECs infected with *T. gondii* early (24 h) also did not exhibit significative changes in the immunological markers analyzed: *iNOS* (Figure 2A), *Arginase* (Figure 2B), *TNF-α* (Figure 2C), *IL-10* (Figure 2D), *IL-1β* (Figure 2E), *TGF-β* (Figure 2F), and *IL-6* (Figure 2G), when comparing the PECs from the resistant BALB/c and susceptible C57BL/6 mice. 

We noticed that rSAG2A alone was not sufficient to modulate *iNOS* (Figure 2A), *TNF-α* (Figure 2C), *IL-1β* (Figure 2E), and *IL-6* (Figure 2G), all genes of pro-inflammatory cytokines. However, rSAG2A alone increased the expression of anti-inflammatory cytokines, *Arginase* (Figure 2B; *p* < 0.01) and *TGF-β* (Figure 2F; *p* < 0.01), while not altering *IL-10* (Figure 2D) expression in the BALB/c compared to C57BL/6 mice. These data indicate an anti-inflammatory property of rSAG2A in resistant BALB/c mice. We next sought to understand if this modulation could affect *T. gondii*-infected PECs. We observed that rSAG2A changed the immune response of the PECs infected with *T. gondii*, but the effects differed between the resistant and susceptible mice. 

First, we note that rSAG2A enhanced the levels of both pro- and anti-inflammatory cytokines genes, *iNOS* (Figure 2A; *p* < 0.001), *TNF-α* (Figure 2C; *p* < 0.001), *IL-10* (Figure 2D; *p* < 0.001), *TGF-β* (Figure 2F; *p* < 0.001), and *IL-6* (Figure 2G; *p* < 0.001) in the C57BL/6 mice. However, the BALB/c mice only presented an increase in anti-inflammatory cytokine genes, *TGF-β* (Figure 2F; *p* < 0.001) and *Arginase* (Figure 2B; *p* < 0.001).

Next, we compared the resistant BALB/c and susceptible C57BL/6 mice treated with rSAG2A and infected with *T. gondii*. Our data showed that the C57BL/6 mice exhibited high expression levels of *iNOS* (Figure 2A; *p* < 0.01), *TNF-α* (Figure 2C; *p* < 0.05), *IL-10* (Figure 2D; *p* < 0.05), and *IL-6* (Figure 2G; *p* < 0.01) compared to the BALB/c mice, indicating a more inflammatory profile. Conversely, the BALB/c mice exhibited high expression levels of *Arginase* (Figure 2B; *p* < 0.001), a marker of alternatively activated macrophages (M2-like). These data indicate that rSAG2A stimulates different profiles in resistant and susceptive mice (Figure 2H) and suggest that the increase in pro-inflammatory cytokine levels helps PECs from C57BL/6 mice to control parasitism (Figure 1). We hypothesized that this effect could be a result of protein accumulation and alterations in the immunological pathways in C57BL/6 mice. To test our hypothesis, we performed a proteomic analysis and a protein–protein interaction network with bioinformatics tools.

### 3.3. rSAG2A Induced Accumulation of Proteins Associated with Oxidative Stress and Immune Pathways in PECs of C57BL/6 Mice

Our results indicated that rSAG2A may act on PECs’ microbicidal mechanisms at the transcription level (Figure 2). To understand the effects of rSAG2A in both the susceptible and resistant mice, we next analyzed the differentially expressed proteins using proteomics. In our analysis, proteins were considered as exclusive to a group based on the obtained FC values: FC > 1 indicates being exclusive for a group, and FC > 2 indicates protein augmentation (see Section 2). 

We identified six exclusive proteins from the C57BL/6 mice infected with *T. gondii*, 16 from the C57BL/6 mice treated with rSAG2A, 12 from the BALB/c mice infected with *T. gondii*, and 42 for the BALB/c mice treated with rSAG2A (Figure 3). Additionally, we observed 50 proteins augmented in the C57BL/6 mice treated with rSAG2A: (i) 15 were shared among BALB/c_rSAG2A, BALB/c_*T. gondii*, and C57BL/6_rSAG2A mice; (ii) 26 were common to all rSAG2A-incubated PECs from both mice lineages; (iii) 5 were shared among the BALB/c mice treated with rSAG2A, BALB/c mice only infected with *T. gondii*, and C57BL/6 mice infected with *T. gondii*; and (iv) 4 proteins were identified as b3int shared between the BALB/c and C57BL/6 mice treated with rSAG2A (Figure 3, indicated with an asterisk). Interestingly, BALB/c mice treated with rSAG2A or infected with *T. gondii* shared 319 common proteins, two-fold higher than C57BL/6 when comparing the treated and infected animals, which shared only 155 proteins. A total of 211 proteins were shared between the BALB/c and C57BL/6 infected mice.

We used hierarchical clustering and FC to identify the proteins exclusively up- or down-modulated in the treated and infected C57BL/6 PECs compared to BALB/c (Figure 4). We identified 66 proteins exclusively up- and 24 exclusively down-modulated in the C57LB/6 PECs treated with rSAG2A compared to the BALB/c treated PECs (Figure 4A). This profile was significantly different from that observed in the C57BL/6 PECs infected with *T. gondii*. Only 37 proteins were exclusively up- and 19 exclusively down-modulated in the C57LB/6 PECs infected with *T. gondii* compared to the BALB/c infected PECs (Figure 4B). These proteins have been related to immune responses or cellular metabolism, such as annexin A7, prostaglandin reductase 1, proteasome subunit alpha type-1, the 26S proteasome non-ATPase regulatory subunit 2, and the hematopoietic lineage cell-specific protein. Among the down-modulated proteins were vinculin, tropomyosin alpha-4 chain, trans-aldolase, and the 40S ribosomal protein S7 (Appendix A).

To understand how rSAG2A affected parasitism in the C57BL/6 PECs, we applied a protein–protein network analysis to identify the pathways and biological processes impacted by rSAG2A and *T. gondii* infection (Figure 5). The integrated network of proteins with a reduced expression from C57BL/6 mice only treated with rSAG2A displayed 760 nodes and 15,850 connectors with a high fidelity. When *T. gondii* was present, 1384 nodes and 41,467 connectors were observed. Cluster analysis of these nodes allowed for the identification of 6 connected protein clusters in the rSAG2A group and 14 clusters when *T. gondii* was present. The main biological processes enriched in these clusters were related to protein metabolism, signaling, oxidative processes like fatty acid beta-oxidation, cell development, chromatin modification, and cytoskeleton organization for rSAG2A (Figure 5A), and glucose metabolism, proton transport, response to stress, cholesterol biosynthesis, and RNA splicing for PECs infected with *T. gondii* (Figure 5B).

The network of exclusive and augmented proteins from the C57BL/6 PECs treated with rSAG2A presented 1391 nodes and 38,537 connectors, as well as 11 clusters (Figure 6A), and when the parasite was present, 1019 nodes and 25,681 connectors were observed (Figure 6B). The biological processes enriched in these clusters were related to (i) actin filament bundle assembly; (ii) the CDP-choline pathway and generation of neurons; (iii) membrane invagination; (iv) heparan sulfate proteoglycan metabolism; (v). the innate immune response; (vi) cell redox homeostasis, oxygen and reactive oxygen species, and the electron transport chain; and (vii) the catabolism of fatty acids, among others, when cells were treated with rSAG2A.

Furthermore, we observed a specific regulation in the intracellular signaling pathways of networks in the C57BL/6 mice treated with rSAG2A, indicating a heightened protein expression of pathways related to the immune response, including the production of intermediate oxygen species. This metabolic profile is typical of an inflammatory immune response and may explain the increased host resistance against *T. gondii* replication in the intracellular environment. Moreover, the presence of other inflammatory proteins in the C57BL/6 mice pretreated with rSAG2A could work together with reactive oxygen species, culminating in pathogen elimination and explaining the reduced intracellular parasitism. In summary, these data indicate that the activation of the studied inflammatory mediators induced by rSAG2A could help to control the infection by *T. gondii* in PECs from susceptible C57BL/6 mice.

### 3.4. Major Genes and Responses Associated with Pro-Inflammatory Cytokines

The protein–protein network interaction associated with cytokines (TNF, IL-6, IL-10, and TGF-β1) and iNOS was analyzed and found to be composed of 173 proteins categorized into five different groups (1 to 5) related to important biological processes (Figure 7). Incubation with rSAG2A showed that the proteins Lgals3, Hsp90ab1, and Mapk1 had direct interactions with TNF, iNOS2, and TGF-β1, respectively. Additionally, the Pdgfrb protein interacted with both IL-6 and IL-10, while Serpinb6c interacted with TNF, IL-6, and IL-10.

The proteins Lgals3, Pdgfrb, and Mapk1 were less abundant when PECs from the C57BL/6 mice were treated with rSAG2A. On the other hand, the levels of the proteins Serpinb6c and Hsp90ab1 were at least two times higher in the same conditions. We also observed a reduction in S100 calcium-binding protein (S100A6). The S100A6 of the host cell is important for the invasion of *T. gondii* by reducing or blocking its functional epitopes induced by parasite invasion [31]. Thus, the reduction in S100A6 could be involved in the decrease in parasite proliferation by the inhibition of parasite invasion. 

In group 5, the proteins that deserve emphasis were those associated with processes such as the production of metabolic precursors and energy, responses to nutrient levels, lipid modifications, the regulation of responses related with receptors, and endocytosis. Among the proteins associated with this group, Hk3, Dld, Phb, Gm10053, Slc25a3, Uqcrc2, Mtch2, Synj1, Ndufv1, Ap2s1, Hsd17b4, Vdac2, and Hk1 were abundant during incubation with rSAG2A. Moreover, the proteins Taldo1, Vdac1, Eno2, Mdh2, Acadm, Cs, and Pgk1-rs7 were present in group 5.

## 4. Discussion

Our knowledge about *T. gondii* surface antigens can shed light on evasion mechanisms, vaccine development, the parasite life cycle, and effective immune response. The literature has shown that surface antigen glycoproteins (SAGs) activate the immune system, and different SAG antigens can exhibit distinct modulatory mechanisms during *T. gondii* infection. For example, recombinant surface antigen SAG1 (rSAG1) reduced fetal infection in resistant mice, but not in susceptible animals, although similar levels of cytokines such as IL-4, IL-10, and IFN-γ were observed in maternal sera during gestation [32]. SAG2A has been previously suggested as a promising candidate to modulate *T. gondii* infection. The activity of macrophages and dendritic cells from C57BL/6 bone marrow stem cells could be modulated by SAG2A, and the immunomodulatory effect of SAG2A has been associated with the C-terminal portion [14]. Our group previously suggested that the rSAG2A, a recombinant protein with the presence of a disordered structure in the C-terminal region, is a promising candidate to modulate *T. gondii* infection [22].

Here, to underscore the effect of rSAG2A in the immune response of mammals, we used resistant (BALB/c) and susceptible (C57BL/6) mouse lineages to collect peritoneal exudate cells (PECs) and treated them with rSAG2A in the presence or absence of *T. gondii* tachyzoites. Our data showed that rSAG2A can induce an anti-inflammatory profile in resistant mice in the absence (increase *Arginase* and *TGF-β*) and presence (increase *Arginase*) of *T. gondii* tachyzoites. Conversely, although rSAG2A alone did not induce a pro-inflammatory profile in the susceptible mice in the absence of *T. gondii* tachyzoites, rSAG2A enhanced the pro-inflammatory profile in the infected PECs, mediated by *iNOS*, *TNF-α*, and *IL-6.* Resistance and susceptibility to *T. gondii* in mouse lineages have been attributed to various factors, including the immune response, T-cell mediated response, and dendritic cell polarization [8,33]. Our data suggest that rSAG2A can enhance the natural response observed in both the C57BL/6 (pro-inflammatory) and BALB/c (anti-inflammatory) lineages. Nonetheless, these modulations in genes involved in the microbicidal mechanisms of macrophages promote the control of *T. gondii* parasitism in C57BL/6 PECs with a low dosage (10 µg/mL^−1^). We hypothesized that the ability of rSAG2A to induce *IL-10* expression in addition to pro-inflammatory cytokines in the C57Bl/6 PECs could help to balance the immune response and avoid extensive tissue damage by controlling excessive inflammation [34].

Macêdo-Junior and colleagues [14] previously demonstrated that a truncated form of the rSAG2AΔ135 protein, lacking its C-terminal end, was also able to induce high levels of iNOS and IL-12 production in a dose-dependent manner in bone-marrow-derived monocytes from C57BL/6 susceptible mice. Furthermore, the immune response is complex, and the microbicidal mechanisms and regulatory mechanisms of macrophages require the activation of different receptors and signaling pathways. To understand if and how rSAG2A controls biological processes and signaling pathways, we used proteomic analysis and bioinformatics tools.

We identified that *T. gondii* infection modulated different exclusive proteins in the resistant and susceptible mice: 6 in C57BL/6 and 12 in BALB/c. However, incubation with rSAG2A increased these exclusive modulations in each model, from 6 to 16 in the C57BL/6 and from 12 to 42 in the BALB/c PECs. As observed by us, this modulation was not restricted to pro- or anti-inflammatory proteins, as suggested in our mRNA expression analysis. In fact, Lee and colleagues [33] previously showed that *T. gondii* infection modulates not only the cytokines in dendritic cells, but also the surface receptors and molecules involved in antigen presentation and T-cell activation, such as major histocompatibility complex (MHC) II, CD40, CD80, and CD86. Among the proteins exclusively modulated, we noticed that the C57BL/6 PECs were more susceptible to rSAG2A and presented more up- and down-modulated proteins than the BALB/c PECs. These up-modulated proteins included Hsp90, aldehyde dehydrogenase (ALDH) 1A, prostaglandins reductase 1, lysozyme type C, and 26S proteasome non-ATPase regulatory subunit 2, among others.

During *T. gondii* infection, Hsp90, a chaperon protein, is down-modulated and helps the parasite to invade cells and enhance tachyzoite transition to the bradyzoite stage, allowing bradyzoites to evade the immune response [35]. We suggested that the enhancement of Hsp90 after treatment with rSAG2A may control parasitism in PECs by controlling bradyzoite formation. Additionally, Sugi and colleagues [36] showed, in a single-cell transcriptome analysis, that the bradyzoites derived from the ME-49 strain had anti-inflammatory activity in human foreskin fibroblasts, down-modulating the NF-ķB and IFN-γ response, which helped to prevent the parasite from being eliminated from the body. 

The activation of cytotoxic mechanisms through the ALDH pathway is common during *T. gondii* infection in both human and mouse cells. We observed that rSAG2A up-modulated not only ALDH1A, but also prostaglandin reductase 1 production in C57BL/6 PECs. Prostaglandin reductase 1 is a key enzyme for the degradation of prostaglandins, which are messenger lipids that regulate processes such as cell survival and inflammation [37]. The increase in lysozyme type C, a family of proteins known to have microbicidal activities, and the up-regulation of the 26S proteasome non-ATPase regulatory subunit 2 and ALDH1A can positively regulate cell-mediated cytotoxicity and the production of reactive oxygen species (ROS) [38,39], thus controlling parasitism. Additionally, we noticed that biological processes crucial to parasite survival and the immune response were modulated by rSAG2A in the C56BL/6 PECs, including fatty acid beta-oxidation, oxygen and reactive oxygen species metabolism, interferon production, and cytokine-mediated signaling pathways. 

## 5. Conclusions

Together, our study not only suggests that recombinant SAG2A controls *T. gondii* parasitism in susceptible C57BL/6 PECs but supports that this control is closely related to modulation in pro-inflammatory cytokines and the enhanced expression of proteins involved in the cytotoxic response and control of the parasitic life cycle. Ou work may also serve as a valuable source for future therapeutic strategies for toxoplasmosis using surface antigen glycoproteins. 

## Figures and Tables

**Figure 1 microorganisms-12-02366-f001:**
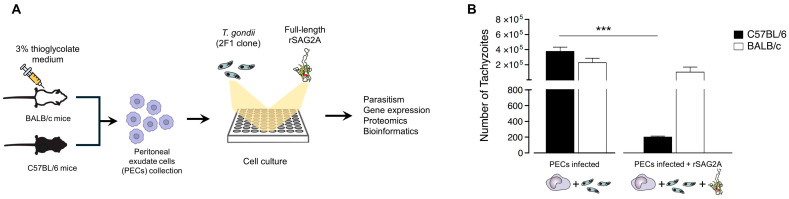
rSAG2A decreases the parasitism in PECs infected with tachyzoites of *T. gondii*. (**A**) PECs from BALB/c and C57BL/6 mice were infected with tachyzoites of *T. gondii* (RH-2FI strain; MOI = 1) for 3 h, followed or not by addition of rSAG2A. (**B**) The number of intracellular tachyzoites was measured by the β -galactosidase colorimetric assay after incubation for 24 h with rSAG2A. Data represent mean + SD of two independent experiments. *** *p* < 0.001 by one-way ANOVA followed by the Tukey post-test.

**Figure 2 microorganisms-12-02366-f002:**
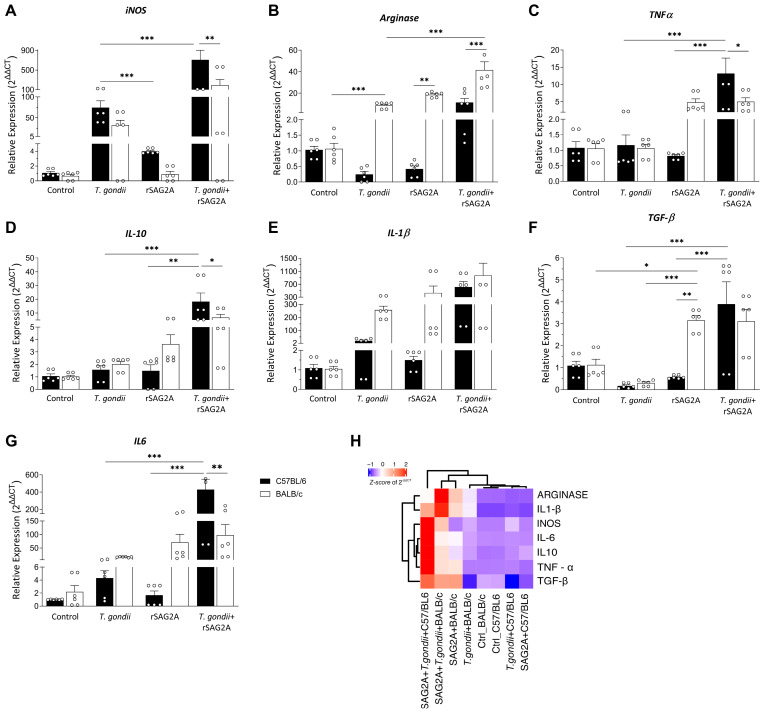
rSAG2A alters the pro-inflammatory and anti-inflammatory cytokines expression in PECs infected with *T. gondii*. Cytokine transcript accumulation was evaluated during the presence of parasites for 3 h in PECs from both BALB/c and C57BL/6 mice or treated with rSAG2A during 24 h, through qPCR. (**A**) INOS, (**B**) Arginase, (**C**) TNF-α, (**D**) IL-10, (**E**) IL-1β, (**F**) TGF-β, and (**G**) IL-6. (**H**) Hierarchical clustering for the same mean relative quantification (RQ) values for the same genes, represented in z score scale, in the different conditions studied. Data represent mean + SD of two independent experiments using a pool of PECs n = 4 animals/experiment. * *p* < 0.05, ** *p* < 0.01, and *** *p* < 0.001 between control and rSAG2A of 6 replicates/group by ANOVA followed by the Tukey post-test.

**Figure 3 microorganisms-12-02366-f003:**
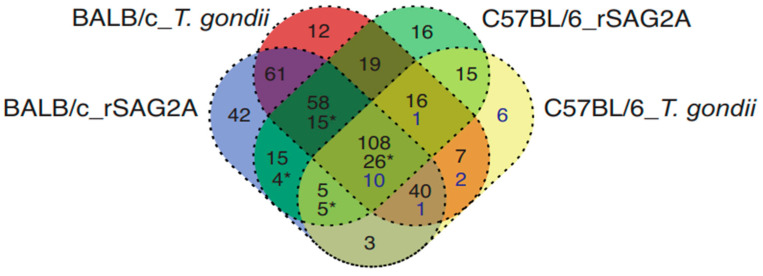
Venn diagram of proteins shared by PECs of mice infected with *T. gondii* or treated with rSAG2A. Venn diagram showing the number of proteins in common and not in common between the different groups. PECs of the BALB/c mice treated with rSAG2A (BALB/c_rSAG2A), BALB/c mice infected with *T. gondii* (BALB/c_*T. gondii*), C57BL/6 mice treated with rSAG2A (C57BL/6_rSAG2A), and C57BL/6 mice infected with *T. gondii* (C57BL/6_*T. gondii*). The asterisk values represent proteins augmented with fold change > 2 exclusively in PECs of C57BL/6 mice treated with rSAG2A.

**Figure 4 microorganisms-12-02366-f004:**
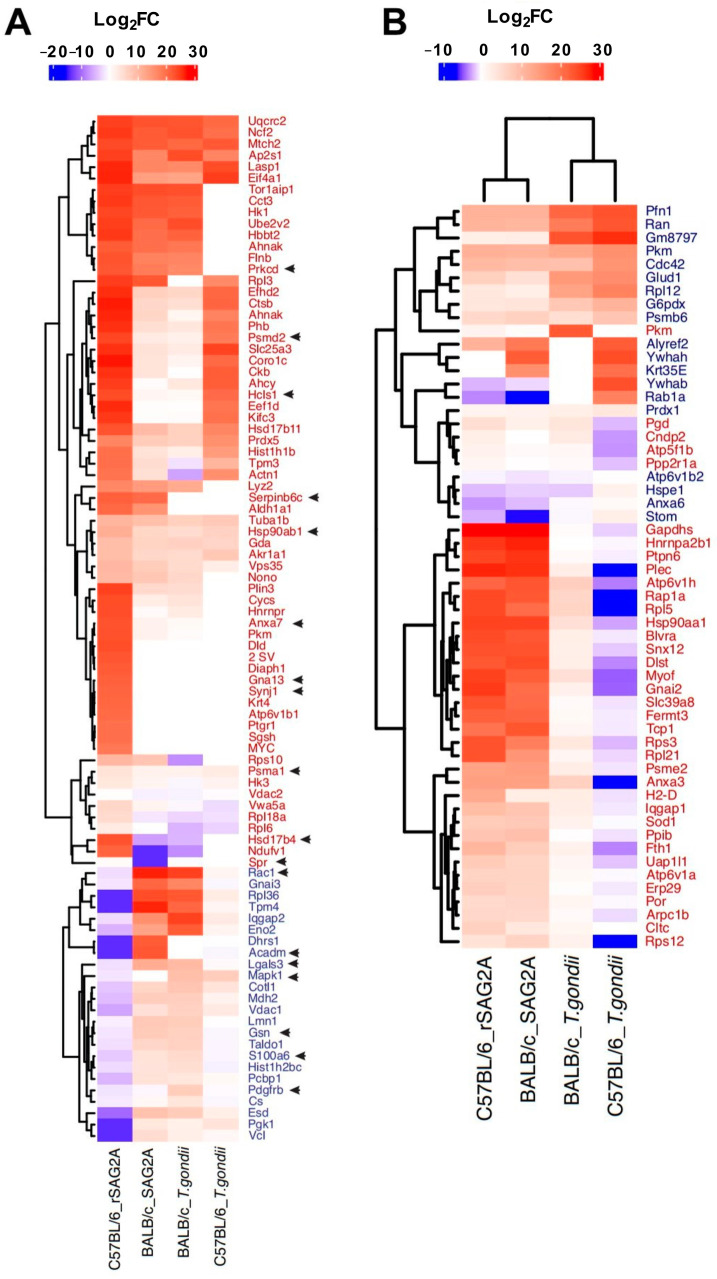
Accumulation profile and hierarchical clustering for exclusive/up-modulated or down-modulated proteins in C57BL/6 mice PECs treated with rSAG2A (**A**) or infected with *T. gondii* (**B**), compared to BALB/c PECs in the same conditions. Proteins in red are those exclusive or that had a Log2FC (compared to the other conditions) at least twice as high in C57BL/6 PECs treated with rSAG2A (red names in **A**) or in C57BL/6 PECs infected with *T. gondii* (red names in **B**). Proteins in blue are those that were repressed only in C57BL/6 PECs treated with rSAG2A (blue names in **A**) or in C57BL/6 PECs infected with *T. gondii* (blue names in **B**). Heatmap colors are represented by Log2FC scale, which indicates the accumulation of proteins in Log2 fold change scale; blue shades indicate the repressed proteins and red shades indicate proteins that increased in relation to their respective controls. Black arrows represent proteins present in the cytokine network.

**Figure 5 microorganisms-12-02366-f005:**
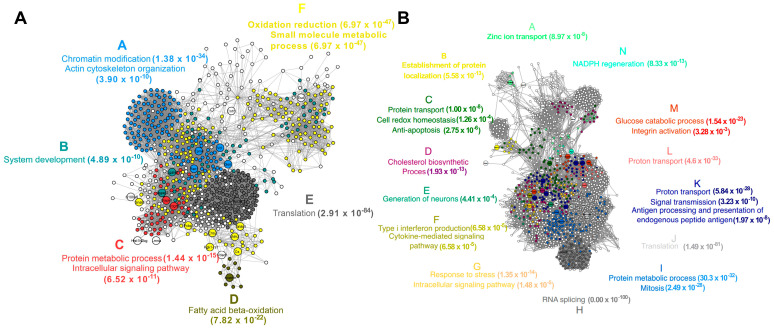
Protein–protein interaction network for exclusive and augmented proteins from PECs of C57BL/6 mice treated with rSAG2A (**A**) or infected with *T. gondii* (**B**). The bigger nodes represent the proteins identified through proteomic analysis. Smaller nodes were aggregated with networks using the STRING database. The different colors represent different clusters identified in the module analysis. The biological processes enriched in each cluster are described with the respective colors around the networks.

**Figure 6 microorganisms-12-02366-f006:**
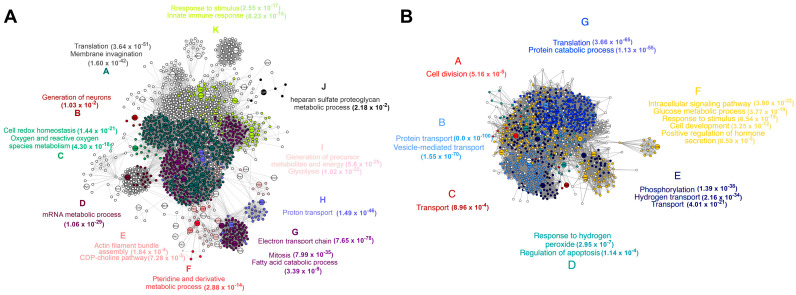
Network of protein–protein interactions for down-modulated proteins in PECs of C57BL/6 mice treated with rSAG2A (**A**) or infected with *T. gondii* (**B**). The bigger nodes represent the proteins identified through proteomic analysis. Smaller nodes were aggregated with networks using the STRING database. The different colors represent different clusters identified in the module analysis. The biological processes enriched in each cluster are described with the respective colors around the networks.

**Figure 7 microorganisms-12-02366-f007:**
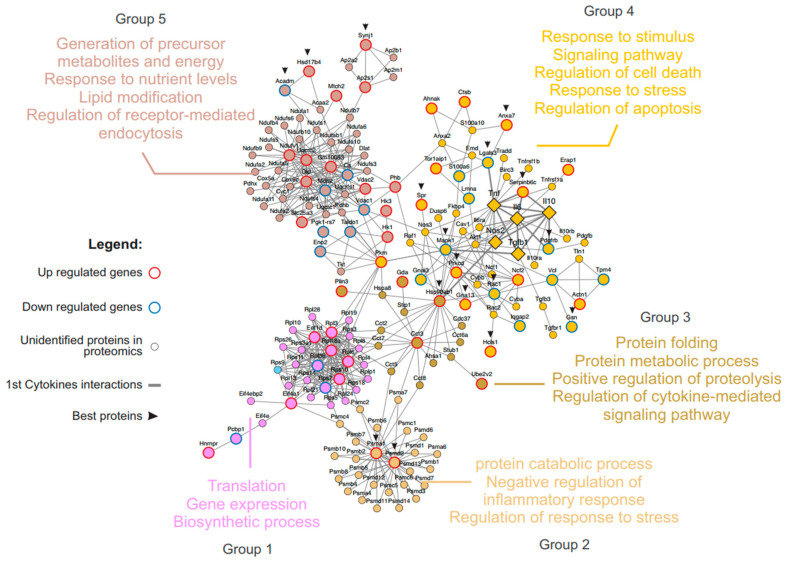
Protein interaction network related with inflammatory and regulatory cytokines. The large circles represent proteins whose abundance was altered in PECs treated with rSAG2A. Diamonds represent the inflammatory and regulatory cytokines, besides INOs. Small circles represent proteins not identified during the proteomic analysis. The fat connectors highlight first-degree interaction with the cytokines and the slender connectors the interactions in distinct degrees. The arrows indicate the most relevant proteins in the study. The red and blue borders represent up-modulated and down-modulated proteins, respectively.

**Table 1 microorganisms-12-02366-t001:** Sequence of primers used in real-time quantitative PCR reactions (RT-qPCR).

Gene	Sequence
*IL-1β*	Forward: 5′-GCACACCCACCCTGCA-3′ Reverse: 5′-ACCGCTTTTCCATCTTCTTCTT-3′
*IL-6*	Forward: 5′-TCCAGAAACCGCTATGAAGTTC-3′ Reverse: 5′-CACCAGCATCAGTCCCAAGA-3′
*IL-10*	Forward: 5′-TGGACAACATACTGCTAACC-3′ Reverse: 5′-GGATCATTTCCGATAAGGCT-3′
*TGF-β*	Forward: 5′-AAGCTACCAAGTTAGACTTCCCA-3′ Reverse: 5′-TGAAAGTTTAGCATACAGAATCCC-3′
*TNF-α*	Forward: 5′-TGTTTCGAGGTTGCTTGTCT-3′ Reverse: 5′-GATTGTTCCACCAGCTTGC-3′
*Arginase1*	Forward: 5′-AAAGCTGGTCTGCTGGAAAA-3′ Reverse: 5′-ACAGACCGTGGGTTCTTCAC-3′
*iNOS*	Forward: 5′-CAGCTGGGCTGTACAAACCTT-3′ Reverse: 5′-CATTGGAAGTGAAGCGTTTCG-3′

## Data Availability

The raw data supporting the conclusions of this article will be made available by the authors on request.

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
