# Peer review of "Recombinant SAG2A Protein from *Toxoplasma gondii* Modulates Immune Profile and Induces Metabolic Changes Associated with Reduced Tachyzoite Infection in Peritoneal Exudate Cells from Susceptible C57BL/6 Mice"

_microorganisms, 2024, doi:10.3390/microorganisms12112366_

Round 1
Reviewer 1 Report
Comments and Suggestions for Authors
The reviewed scientific work presents a high level of molecular and proteomic research on selected proteins in host cells that play an important role in the course of Toxoplasma gondii invasion. This is a contribution to understanding the mechanisms related to the pathogenesis of invasion and the mechanisms of the immune response. It concerns in vitro analysis that reflects the reactions occurring in the cells of the infected organism.
I have no substantive comments regarding the course of the experience. It was ignited and performed in accordance with molecular and proteomic research standards. However, I have a few editorial comments. The authors use the term treatment in the context of the analyzed cells. In my opinion, this term can only refer to an infected living organism and refers to a disease, not an invasion. In this scientific work, no in vivo studies were conducted, therefore the term "treatment" should not be used. Moreover, it is incomprehensible why the characteristics of the parasite with details of the development cycle were not presented in the introduction. An additional advantage of the work would be a detailed description of the development of the intermediate host, with an acute phase associated with tachyzoites and a chronic phase associated with bradyzoites.
in line 43 it is not explained what population is meant. In addition to humans, toxoplasmosis is a pathogen in many other host species. Verse 57 - cysts or oocysts do not penetrate tissue barriers - this applies to trophozoites, which, depending on the phase of invasion, are sporozoites, tachyzoites or bradyzoites. In general, the work is very good, but it lacks a parasitologist's perspective. Therefore, I suggest supplementing this article with descriptions of the course of the invasion, selected fragments of which are tested in vitro.
Author Response
Comment 1: The reviewed scientific work presents a high level of molecular and proteomic research on selected proteins in host cells that play an important role in the course of Toxoplasma gondii invasion. This is a contribution to understanding the mechanisms related to the pathogenesis of invasion and the mechanisms of the immune response. It concerns in vitro analysis that reflects the reactions occurring in the cells of the infected organism.
I have no substantive comments regarding the course of the experience. It was ignited and performed in accordance with molecular and proteomic research standards. However, I have a few editorial comments. The authors use the term treatment in the context of the analyzed cells. In my opinion, this term can only refer to an infected living organism and refers to a disease, not an invasion. In this scientific work, no in vivo studies were conducted, therefore the term "treatment" should not be used. Moreover, it is incomprehensible why the characteristics of the parasite with details of the development cycle were not presented in the introduction. An additional advantage of the work would be a detailed description of the development of the intermediate host, with an acute phase associated with tachyzoites and a chronic phase associated with bradyzoites.
Response 1: We thank Reviewer 1 for the valuable comments. We agree that the introduction needs more information regarding parasitic life cycle. As suggested, we removed the term “treatment” throughout the manuscript as suggested. Additionally, we added more information about the parasite in the Introduction section.
Comment 2: in line 43 it is not explained what population is meant. In addition to humans, toxoplasmosis is a pathogen in many other host species. Verse 57 - cysts or oocysts do not penetrate tissue barriers - this applies to trophozoites, which, depending on the phase of invasion, are sporozoites, tachyzoites or bradyzoites. In general, the work is very good, but it lacks a parasitologist's perspective. Therefore, I suggest supplementing this article with descriptions of the course of the invasion, selected fragments of which are tested in vitro.
Author’s response: We have added the term “humans” to clarify the population cited. We apologized to the reviewers and have corrected the phrase regarding cysts and oocysts. We also added more information about the parasite in the Introduction section.
Reviewer 2 Report
Comments and Suggestions for Authors
This study investigates recombinant SAG2A protein's impact on immune response and metabolic changes in Toxoplasma gondii infection, showing reduced parasitism and modulated pro-inflammatory cytokines in susceptible C57BL/6 mice, highlighting SAG2A's potential as a therapeutic candidate for controlling T. gondii through immune modulation. The experiments were well established and designed. The data presented could support the conclusion. I only have a few minor comments for the author to address.
Line 52-54 how about dense granule proteins?
Use the italic form for all T. gondii appeared in this manuscript.
Line 147: "GO Taq qPCR Master Mix" should be "GoTaq qPCR Master Mix".
Line 256: "T-student post-test" should be "t-test".
Line 275: please specify One-way or Two-way ANOVA.
Line 375: "Hierachical" should be "Hierarchical".
Line 522: "colesly related" should be "closely related".
Line 524: "lyfe cicle" should be "life cycle".
Line 525: "terapeutic" should be "therapeutic".
Author Response
Comment 1: This study investigates recombinant SAG2A protein's impact on immune response and metabolic changes in Toxoplasma gondii infection, showing reduced parasitism and modulated pro-inflammatory cytokines in susceptible C57BL/6 mice, highlighting SAG2A's potential as a therapeutic candidate for controlling T. gondii through immune modulation. The experiments were well established and designed. The data presented could support the conclusion. I only have a few minor comments for the author to address.
Response 1: We thank the Reviewer 2 for the valuable comments to improve the quality of our manuscript.
Comment 2: Line 52-54 how about dense granule proteins?
Response 2: We agree with the reviewer and have included a new mention to dense granule antigens (GRA).
Comment 3: Use the italic form for all T. gondii appeared in this manuscript.
Response 3: We have corrected mention of T. gondii throughout the manuscript, ensuring it is in italic form.
Comment 4: Line 147: "GO Taq qPCR Master Mix" should be "GoTaq qPCR Master Mix".
Response 4: We have corrected the spelling mistake.
Comment 5: Line 256: "T-student post-test" should be "t-test".
Response 5: We have corrected the spelling mistake.
Comment 6: Line 275: please specify One-way or Two-way ANOVA.
Response 6: We have added the required information: One-way ANOVA.
Comment 7: Line 375: "Hierachical" should be "Hierarchical".
Response 7: We have corrected the spelling mistake.
Comment 8: Line 522: "colesly related" should be "closely related".
Response 8: We have corrected the spelling mistake.
Comment 9: Line 524: "lyfe cicle" should be "life cycle".
Response 9: We have corrected the spelling mistake.
Comment 10: Line 525: "terapeutic" should be "therapeutic".
Response 10: We have corrected the spelling mistake.